# Off-Beat Multi-Agent Reinforcement Learning

## Abstract

We investigate model-free multi-agent reinforcement learning (MARL) in environments where *off-beat* actions are prevalent, *i.e.*, all actions have pre-set execution durations. During execution durations, the environment changes are influenced by, but not synchronised with, action execution. Such a setting is ubiquitous in many real-world problems. However, most MARL methods assume actions are executed immediately after inference, which is often unrealistic and can lead to catastrophic failure for multi-agent coordination with off-beat actions. In order to fill this gap, we develop an algorithmic framework for MARL with off-beat actions. We then propose a novel episodic memory, LeGEM, for model-free MARL algorithms. LeGEM builds agents' episodic memories by utilizing agents' individual experiences. It boosts multi-agent learning by addressing the challenging temporal credit assignment problem raised by the off-beat actions via our novel reward redistribution scheme, alleviating the issue of non-Markovian reward. We evaluate LeGEM on various multi-agent scenarios with off-beat actions, including Stag-Hunter Game, Quarry Game, Afforestation Game, and StarCraft II micromanagement tasks. Empirical results show that LeGEM significantly boosts multi-agent coordination and achieves leading performance and improved sample efficiency.

## 1  Introduction

In Multi-Agent Reinforcement Learning (MARL), multiple agents act interactively and complete tasks in a sequential decision-making manner with Reinforcement Learning (RL). It has made remarkable advances in many domains, including autonomous systems [8, 19, 72] and real-time strategy (RTS) video games [58]. By the virtue of the *centralised training with decentralised execution* (CTDE) [33] paradigm, which aims to tackle the scalability and partial observability challenges in MARL, many CTDE-based MARL methods are proposed [13, 49, 41, 62, 47, 63, 23, 35]. With these methods, an agent executes actions only via feeding its individual observations independently and optimizes its own policy with access to global trajectories centrally.

Despite the recent successes of MARL, learning effective multi-agent coordination policies for complex multi-agent systems remains challenging. One key challenge is the *off-beat* actions, *i.e.*, all actions have pre-set execution durations[1] and during the execution durations, the environment changes are influenced by, but not synchronised with, action execution (an illustrative scenario is shown in Fig. 1). However, Dec-POMDP [32], which underpins many CTDE-based MARL methods, hinges on the assumption that actions are executed momentarily after inference, leading to catastrophic failure for *centralized training* on various off-beat multi-agent scenarios (OBMAS). To fill this gap, we study MARL in settings where off-beat actions are prevalent. Such setting is very common in many real-world problems. For example, in the traffic light control problem, traffic lights in the conjunctions of the road network have pre-set execution time which is set asynchronously.

---

[1]In the RL literature [39, 6], action execution durations are called *delays of actions*. In this paper, we use the term *execution durations*, which is self-consistent with off-beat actions defined in Sec. 3.

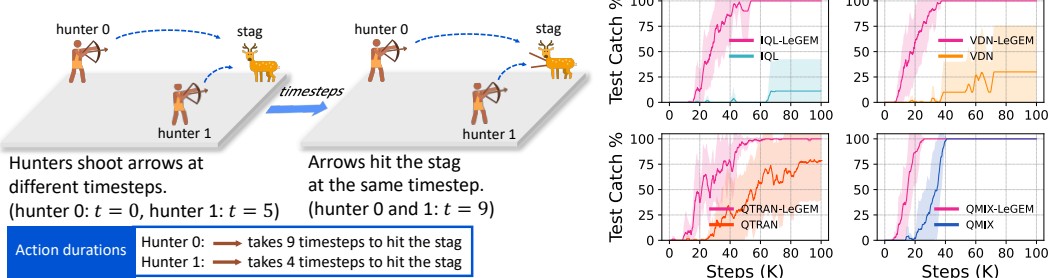

Figure 1: **An illustrative scenario:** two-agent stag-hunter game, where two agents (hunters) have only partial observations, different durations of the shoot action, and cannot communicate. The goal is to catch the stag and they are rewarded when their shot hits – as in, completion of the action is synchronised, the stag at the same time. Both agents can see the stag. As the shoot action durations of the two agents are different, to catch the stag, the two agents should shoot the arrow at different timesteps given the distances. Though the scenario is easy for human beings, it is hard for MARL agents due to the action duration. **Experiment results:** in this scenario, the optimal policy for agent 0 is to shoot the arrow at timestep 0 while the optimal policy for agent 1 is to shoot the arrow at timestep 5. Such asynchronous property of OBMAS motivates agents to learn tacit policies. The curves show that VDN and IQL fail to learn coordination policies even in this simple scenario. With LeGEM, MARL methods gain enhanced performances as well as improved sample efficiency.

The problem of off-beat actions in MARL has yet to be investigated and tackled. Training MARL policies in OBMAS is challenging: (i) Each agent's actions can have a variety of execution durations, which augments the order of complexity of OBMAS during decentralized execution, resulting in failure of the coordination; (ii) The action durations are unknown to agents during individual executions, and communication is constrained and not always feasible, making it non-trivial to model the environment; (iii) During training, both the temporal credit assignment with TD-learning [51] and the *inter-agent* credit assignment with value decomposition methods [41] cannot perform well due to the displaced rewards in multi-agent replay. With off-beat actions, the nonstationarity issue, which mainly stems from rewards' time dependency on the agents' past actions, is exacerbated.

While actions durations are ubiquitous, existing works only focus on single-agent settings, *i.e.*, delay, in RL. Many approaches [59, 39, 66] augment the state space with the queuing actions to be executed into the environment. However, such state-augmentation trick leads to exponentially increasing training samples with the growing action duration, making training intractable [11]. Chen et al. [10] extend the delayed MDP [39] and propose Delayed Markov Game for MARL. However, on one hand, such state-augmentation treatment is confined to short delays, *e.g.*, one timestep delay; on the other hand, the delayed timestep of the actions is privileged information, which is not available in many scenarios. Recent works on macro-actions [67, 68] introduce asynchronous actions by designing macro-actions with prior environment knowledge. Macro-actions are different from options in hierarchical RL (HRL) [52, 3] in that the later is not manually designed but learned. The key difference between macro-actions and off-beat actions is that macro-actions are high-level actions while off-beat actions are primitive actions. Unfortunately, the *inter-agent* credit assignment is still a challenge of HRL in OBMAS and the asynchronous [2] nature of off-beat actions undermines the temporal credit assignment of *centralized training*, causing poor sample efficiency and unsatisfactory performance (more discussions can be found in the related works section in Sec. 7).

We aim to address the aforementioned issues. We first propose off-beat Dec-POMDP. We then instantiate a new class of episodic memory, LeGEM, for model-free MARL algorithms. LeGEM boosts multi-agent learning by addressing the challenging temporal credit assignment problem raised by the off-beat actions via our novel levelled graph-based temporal recency reward redistribution scheme. Specifically, each agent maintains LeGEM and during centralized training, each agent searches the pivot timestep given observations from its graph. The pivot timestep is the timestep wherein the off-beat reward relates to the given node. The pivot timesteps of each agent are ranked, in which the final pivot timestep will be chosen by recency and later used for reward redistribution and target estimation in TD-learning. We evaluate our method on Stag-Hunter Game, Quarry Game, Afforestation Game, and StarCraft II micromanagement tasks. Empirical results show that our method significantly boosts multi-agent coordination and achieves leading performance as well as improved sample efficiency.

---

[2]We clarify the term *asynchronous*: actions that simultaneously committed into the environment by all agents in MARL will not complete their respective action durations at the same time in future timesteps.

## 2 Preliminaries

**Dec-POMDP.** A cooperative MARL problem can be modeled as a *decentralised partially observable Markov decision process* (Dec-POMDP) which can be formulated as a tuple $\langle \mathcal{S}, \mathcal{U}, \mathcal{P}, R, O, \mathcal{N}, \gamma \rangle$, where $s \in \mathcal{S}$ denotes the state of the environment. Each agent $i \in \mathcal{N} := \{1, ..., N\}$ chooses an action $u_i \in \mathcal{U}$ at each timestep, forming a joint action vector, $\boldsymbol{u} := [u_i]_{i=1}^N \in \mathcal{U}^N$. The Markovian transition function can be defined as $\mathcal{P}(\boldsymbol{s}'|\boldsymbol{s}, \boldsymbol{u}) : \mathcal{S} \times \mathcal{U}^N \times \mathcal{S} \mapsto [0, 1]$, transiting one state of current timestep to the state of next timestep conditioned on current state and joint action. Every agent shares the reward and the reward function is $R(\boldsymbol{s}, \boldsymbol{u}) : \mathcal{S} \times \mathcal{U}^N \mapsto \mathcal{R}$. $\gamma \in [0, 1)$ is the discount factor. Due to *partial observability*, each agent has individual partial observation $o \in \mathcal{O}$, according to the observation function $O(\boldsymbol{s}, i) : \mathcal{S} \times \mathcal{N} \mapsto \mathcal{O}$. The goal of each agent is to optimize its own policy $\pi_i(u_i|\tau_i) : \mathcal{T} \times \mathcal{U} \mapsto [0, 1]$ given its action-observation-reward history $\tau_i \in \mathcal{T} := (\mathcal{O} \times \mathcal{U})$.

**Multi-Agent Reinforcement Learning.** MARL aims to learn optimal policies for all the agents in the team. With TD-learning and a global Q value proxy $Q^{\text{tot}}$ for the optimal $Q^*$, $\{Q_i\}_{i=1}^N$ are optimized via minimizing the loss [65, 31]: $\theta^* = \arg\min_{\theta^*} \mathcal{L}(\theta) := \mathbb{E}_{D' \sim \mathcal{D}}[(y_t^{\text{tot}} - Q_\theta^{\text{tot}}(\boldsymbol{s}_t, \boldsymbol{u}_t))^2]$, where $y_t^{\text{tot}} = r_t + \gamma \max_{\boldsymbol{u}'} Q_{\bar{\theta}}^{\text{tot}}(\boldsymbol{s}_{t+1}, \boldsymbol{u}')$ and $\theta$ is the parameters of the agents. $\bar{\theta}$ is the parameter of the target $Q^{\text{tot}}$ and is periodically copied from $\theta$. $D'$ is a sample from the replay buffer $\mathcal{D}$.

## 3 Off-Beat Dec-POMDP

We introduce our formulation for OBMAS. We first define the off-beat actions[3] for multi-agent scenarios; then we propose the Off-Beat Dec-POMDP. All the proofs can be found in Appx. A.

---

**Definition 1** (Off-Beat Actions). Off-beat action $\tilde{u} \in \mathcal{U}$ characterizes OBMAS where the action $\tilde{u}_i$ taken by agent $i$ has execution duration $m_{\tilde{u}_i} \sim A(m|\tilde{u}_i, i)$, $A \in \mathcal{A}$, $m \in \{0, 1, 2, \cdots, M\}$ and $M \leq T$, where $T$ is the maximum duration and $A$ is the action duration distribution. It is a distribution and takes $\tilde{u}_i$ and the index of the agent as parameters. $A$ can be either stochastic or deterministic. The joint off-beat action is $\tilde{\boldsymbol{u}} = [\tilde{u}_i]_{i=1}^N$. The execution duration is decided at the time the action was committed to the environment. Thus, the execution duration of an action $\tilde{\boldsymbol{u}}_t$ initiated at timestep $t$ is $\boldsymbol{m}_t = \{m_{\tilde{u}_i^t}\}_{i=1}^N$.

---

Note that for each agent, $m_{\tilde{u}_i^t}$[4] can be different. At timestep $t$, there are at least 1 action [5] and at most $N$ actions being initiated (committed to the environment for execution), leading to asynchronicity of the joint actions. Next, we propose the Off-Beat Dec-POMDP for OBMAS and discuss its properties.

---

**Definition 2** (Off-Beat Dec-POMDP). Off-Beat Dec-POMDP extends Dec-POMDP, such that (1) state space is $\mathcal{S}$; (2) joint action space is $\mathcal{U}^N$; (3) action duration space is $\mathcal{A}^N$;
(4) transition function is $\mathcal{P}(\boldsymbol{s}'|\boldsymbol{s}, \tilde{\boldsymbol{u}}, \boldsymbol{m}) : \mathcal{S} \times \mathcal{U}^N \times \mathcal{S} \times \mathcal{A}^N \mapsto [0, 1]$, and $\boldsymbol{m}$ is the action durations of the joint action;
(5) the reward function is $R(\boldsymbol{s}, \tilde{\boldsymbol{u}}, \boldsymbol{m}) : \mathcal{S} \times \mathcal{U}^N \times \mathcal{A}^N \mapsto \mathcal{R}$;
(6) we call a reward $r$ as off-beat reward when any its $m_{\tilde{u}_i} \geq 1$, $m_{\tilde{u}_i} \in \boldsymbol{m}$, and $r \neq 0$.

---

In OBMAS, at each timestep $t$, the environment receives actions that agent initiates for execution in the environment. The initiated actions $\tilde{\boldsymbol{u}}_t$ are instantaneous actions inferred by agents' policies given individuals' observations. The joint reward is the consequence of the committed joint actions of current timestep and previous timesteps, depending on the actions' duration. The asynchronicity is an inherent feature of the environment, which is different from asynchronicity incurred by communication delays in many video games (asynchronous gameplay[6]). We discuss some properties of Off-Beat Dec-POMDP below.

---

[3]Asynchronicity is prevalent in real-world multi-agent scenarios, including asynchronicity in observations, actions and communication, etc. In this paper, we focus on the asynchronicity of actions in multi-agent scenarios. For brevity, we name the asynchronicity of actions in MARL as *off-beat*.

[4]We will omit $t$ in the rest of the paper for brevity.

[5]We note that agents have a special NO-OP action available.

[6]https://www.whatgamesare.com/2011/08/synchronous-or-asynchronous-definitions.html

**Remark 1.** *When the durations for all actions are identical, off-beat Dec-POMDP reduces to Delayed Dec-POMDP and there is no off-beat actions in it.*

**Remark 2.** *There exists $\tilde{\boldsymbol{u}}$ that is synchronous since duration of agents' actions can be $m = 0$. When $m$ of all actions is $zero$, off-beat Dec-POMDP reduces to Dec-POMDP.*

In Delayed Dec-POMDP, actions have the same delayed timesteps, which is different from off-beat actions where actions have different action durations or delays. In order to investigate the problem, we consider the deterministic setting of the transition function and the reward function.

**Remark 3** (Non-episodic Reward). *In our formulation, the reward is not episodic reward [16].*

**Remark 4** (Non-Markovian Reward). *With off-beat actions, the Markovian property of the reward function $R(\boldsymbol{s}, \tilde{\boldsymbol{u}}, \boldsymbol{m})$ does not hold.*

With off-beat actions, the shared rewards can be readily displaced, causing non-Markovian rewards. Solving Off-Beat Dec-POMDP is challenging as discussed in Sec. 1. We propose our methods to tackle aforementioned challenges.

# 4 The Journey is the Reward: A Collective Mental Time Travel Method

We propose two methodological elements for Off-Beat MARL. The first, LeGEM, presented in this section, is a form of episodic memory that facilitates discovery of a pivotal timestep for off-beat rewards; and the second, presented in Sec. 5, is redistribution of the off-beat reward to the pivot timestep when the relevant off-beat actions were initialised.

## 4.1 LeGEM: A Levelled Graph Episodic Memory for Off-Beat MARL

Human learning relies on retrospecting our detailed memory of the past [55, 48]. For example, while exploring a new scenic area, we do not just remember a multitude of specific spots there, but can recall the paths that connect them with junctions and turns. However, there is no MARL method that can explicitly recall the past and identify key states that lead to future rewards. Such "*mental time travel*" [24] ability is vital for tackling the challenges in OBMAS. Inspired by the recent progress in RL with episodic memory [18, 5, 17] that is based on the memory prosthesis proposed by neuroscientists [55, 48], we propose our method of episodic memory representation for MARL. Unlike previous episodic memory methods that train a parameterized memory by either augmenting the policy inputs for execution [18] or regularizing the TD learning [17] for RL, our method utilizes the levelled graph data structure [4], a well established structure for data storage and retrieval, to represent an agent's individual episodic memory.

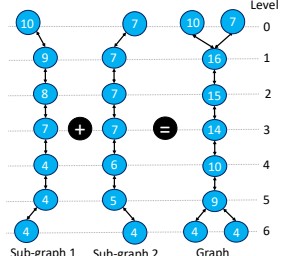

We propose our novel episodic memory, Levelled Graph Episodic Memory method (LeGEM), via the levelled graph data structure. LeGEM memorizes each agent's past trajectories which are partial observations and the unilateral action of the agent. During training, each agent $i$ collects its individual trajectories $\tau_i$. We then define $\tau_i$ of agent $i$ as $\tau_i = [(o_i^0, \tilde{u}_i^0, r^0), \cdots, (o_i^{T-1}, \tilde{u}_i^{T-1}, r^{T-1})]$, where $T$ is the length of the trajectory and the triplet $(o_i^t, \tilde{u}_i^t, r^t)$ represents the observation, action and reward of timestep $t$. Note that $r^t$ is globally shared between agents. We define agent $i$'s LeGEM as a directed graph $\phi_i^t \in \Phi_i$ where $\Phi_i$ is the set of graphs of agent $i$ and $\phi_i^t$ is the $t$-th graph of $\Phi_i$, $t \in \{0, \cdots, T-1\}$. Each $\phi_i^t$ consists of a tuple of $(\Psi, \Xi)$ where $\Psi$ is the set of nodes and $\Xi$ represents the set of edges that connect nodes in the graph. To model an agent's behaviour explicitly and make the trajectories of agents easy to represent, we create $T$ graphs for each agent and let $\Phi_i = \{\phi_i^t\}_{t=0}^{T-1}$ where $T$ is the maximum level of all graphs and the maximum length of the episode as well. The maximum level of $\phi_i^t$ is $t + 1$. The node contains key, visit count and pointers connecting the precursors (node at the previous level) and the successors (node at the next level). Unlike many parameterized episodic memory using state/observation as the key [18, 24], we resort to *afterstate* [36]. That is, we use agent $i$'s observation $o_i^t$ and action at timestep $t$, $\tilde{u}_i^t$, to define the key $(o_i^t, \tilde{u}_i^t)$. We provide an example to showcase the relationship between sub-graph and the graph in Fig. 2. For complex and continuous state scenarios, for example StarCraft II scenarios, we

Figure 2: The maximum level of the graph is 7. Circles indicate the nodes and numbers indicate the visit count.

**Algorithm 1:** `SearchPivotTimesteps` $(\rho)$

1 **Input:** $\tau$, $\Phi$, $\Upsilon$ and `Search` (scheme I or II);
2 **Initialize:** $\kappa$: an empty list to store pivot timesteps;
   // Length of $\tau$ and $\tau_i$ are equal.
3 $l \leftarrow$ `length`$(\tau_i)$-1;
4 **for** $t \leftarrow 0$ **to** `length`$(\tau)$-1 **do**
5    **if** $r^t \neq 0$ $(r^t \in \tau)$ **then**
        // Off-beat reward
6      **for** $i \leftarrow 1$ **to** $N$ **do**
7        Get $\tau_i$ from $\tau$;
8        $\phi_i^l \leftarrow \Phi_i[l]$;
9        $\psi \leftarrow \phi_i^l.$`getNode`$(o_i^t, \tilde{u}_i^t)$;
10        Find all the paths $\Lambda_i^{t,l}$ from node $\psi$ to the node at level 0;
11        Get the discretized episode return $\boldsymbol{r}^{l,i}$;
12        Get the index $\omega$ from $\Upsilon$ with $\boldsymbol{r}^{l,i}$;
13        $e_t^i \leftarrow$ `Search`$(\omega, \Lambda_i^t, \tau_i, \boldsymbol{r}^{l,i}, \Upsilon, \Phi_i)$;
14      Get $e_t$ (Eqn. 1) and append $e_t$ to $\kappa$;
15 **Return:** $\kappa$.

**Algorithm 2:** Search Scheme I

1 **Input:** $\omega$, $\Lambda_i^{t,l}$, $\tau_i$, $\boldsymbol{r}^{l,i}$, $\Upsilon$ and $\Phi_i$;
2 **Initialize:** $\boldsymbol{e}_t^i$: a list whose values are all $t$ and its size is the number of paths in $\Lambda_i^{t,l}$;
3 $\phi_i^{l,\omega} \leftarrow \Phi_i^{l,\Omega}[\omega]$;
4 vc $\leftarrow$ `VisitCount`$(\Lambda_i^{t,l})$ (Alg. 4);
5 **foreach** *path* $\Lambda_i^{t,l}[j] \in \Lambda_i^{t,l}$ **do**
6    $e_t^{i,j,\downarrow} \leftarrow$ `UL`$(\Lambda_i^{t,l}[j], \text{vc}, \tau_i)$ (Alg. 5);
7    $e_t^{i,j,\uparrow} \leftarrow$ `LU`$(\Lambda_i^{t,l}[j], \text{vc}, \tau_i)$ (Alg. 6);
8    **if** $e_t^{i,j,\downarrow} \neq -1$ **then**
9      $\boldsymbol{e}_t^i[j] \leftarrow e_t^{i,j,\downarrow}$;
10    **else if** $e_t^{i,j,\uparrow} \neq -1$ **then**
11      $\boldsymbol{e}_t^i[j] \leftarrow e_t^{i,j,\uparrow}$;
12    **else**
13      $\boldsymbol{e}_t^i[j] \leftarrow t$;
14 $\boldsymbol{e}_t^i \leftarrow$ `Summarize`$(\boldsymbol{e}_t^i)$ (Alg. 7) ;
15 **Return:** $\boldsymbol{e}_t^i$.

use SimHash [9] to discretize the key $(o_i^t, \tilde{u}_i^t)$. This technique has been widely used in commercial search engines and RL [54]. Visit count indicates the total visits made by agent $i$ to the node. It initial value is 1. Note that nodes are bidirectional since it is helpful for searching (see Sec. 4.2).

Given a $\tau_i$ with the length of $T$, if the node is already in the graph at level $t$, we then increase the visit count by 1. Otherwise, we create a new node for level $t$ of the graph and update its pointers. Meanwhile, sub-graphs will be also created and updated. The process of updating LeGEM is in Alg. 3. We provide an example of Alg. 3 in Fig. 9, Appx. B.1. It is worth noting that $\tau_i$ is generated via the interaction of the agent with the environment, and there is no extra interaction needed to collect $\tau_i$. The generated trajectories are saved in the experience replay and later sampled for MARL training.

### 4.2 Multi-Agent Collective Mental Time Travel with LeGEM

With structured agent's past experiences, it can be used to search the pivot timestep when actions that triggered the rewarded state were executed. For example, with LeGEM, we can find the pivot timestep, $e_t = 5$, when agent 1 shoots the arrow in Fig. 1.

**Fact 1.** (Action-Reward Association) *When an off-beat reward $r_t$ exists in the trajectory $\tau_i$ ($i \in \{1, \cdots, N\}$), $r_t \in \tau_i$, off-beat action $\boldsymbol{u}_{t'}$ exists ($t' < t$) in the trajectory set $\{\tau_j\}_{j=1}^N$, where $\{\tau_j\}_{j=1}^N$ constitutes the global trajectory of all agents.*

As the reward function and transition function are deterministic in our setting, Fact 1 holds. Intuitively, once we find an off-beat reward in a trajectory, we are sure that the action which triggered the reward can be found in the trajectory. With more experiences collected by the agents, such pattern is obvious and significant. It motivates us to propose a method to leverage the association property of the off-beat action-reward data and search the pivot timestep for timesteps when off-beat rewards occur, which can further help to redistribute the reward backward to mitigate the temporal credit assignment issue (c.f. Sec. 5). Therefore, we first propose a search method to search the pivot timestep and then propose a proximal ranking method to estimate the pivot timestep that invokes the future reward.

**Collective Mental Time Travel.** The displaced rewards in the replay buffer hinder multi-agent learning. It is essential for each agent to search the pivot timestep when the potential off-beat action that triggered the rewarded state was committed to the environment. Therefore, we propose two search schemes to find the pivot timestep for all agents given an off-beat reward.

*Scheme* I: For agent $i$, given $r_t \in \tau_i$, episode return $\boldsymbol{r}^{l,i}$ of $\tau_i$, $\phi_i^l = \Phi_i[l]$ and $\phi_i^{l,\omega} = \Phi_i^{l,\Omega}[\omega]$ , agent $i$ searches from the node (the key is $(o_i^t, \tilde{u}_i^t)$ and $o_i^t \in \tau_i$, $u_i^t \in \tau_i$) at level $t$ in sub-graph $\phi_i^{l,\omega}$ to find the pivot timestep $e_t$ for $r_t$. Concretely, we propose our bi-directional search method. The first one is called Low-Up (LU) search, which traverses from the given node at level $t$ upwards to the node at level 0. The second one is named Up-Low (UL) search which traverses from the node at level 0 downwards to the given node at level $t$. LU traversing ends when the pattern of increasing visit count

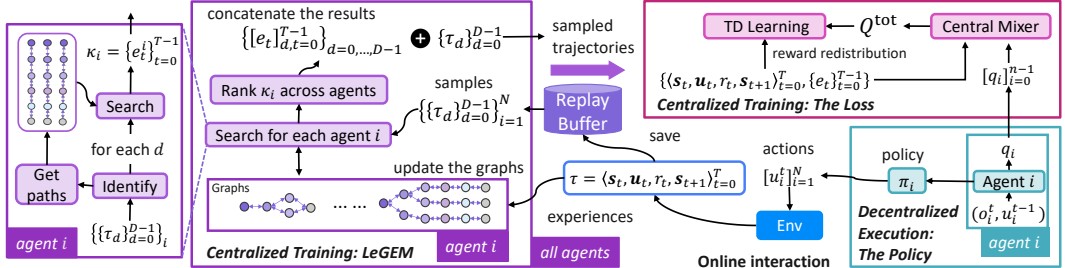

Figure 3: Our framework: LeGEM, the loss and the agent's policy.

ends and the corresponding level is the candidate pivot timestep. On the contrary, UL traversing ends when the pattern of decreasing visit count ends and the corresponding level is the candidate pivot timestep. In Alg. 2, we first get visit count (Line 4) and then apply UL traversing (Line 6) and LU traversing (Line 7). We summarize the results (Line 14) by select the pivot timestep that has the maximum count. UL traversing has a higher searching priority than its counterpart. The reason is that there exists pattern that the visit count is decreasing from the node at level $0$ and such pattern ends at the pivot timestep. In practise, it works well in scenarios whose trajectories are single-off-beat-reward trajectories (there is only one off-beat reward) and the accuracy of Scheme I is over 90% in grid world scenarios. For scenarios, especially complex scenarios, whose trajectories are multiple-off-beat-reward trajectories, we apply Scheme II. We put Alg. 4, Alg. 5, Alg. 6 and Alg. 7 in Appx. B.1 as these algorithms are intuitive and easy to understand literally. The time complexity is $\mathcal{O}(n \cdot m)$ (a slight notation abuse) where $n$ is the size of each $\Lambda_i^{t,l}$ and $m$ ($1 \leq m \leq n$) is the average distance between the level of the given node to the level of the node at the pivot timestep.

*Scheme* II: Scheme II is a simplified version of scheme I for scenarios that have multiple-off-beat-reward trajectories, which searches the pivot timestep by finding the nearest timestep in the most visited path. The node of the nearest timestep has the maximum visitcount in that path. Despite the simplicity, it works effective and the time complexity is $\mathcal{O}(n)$ where $n$ is the number of paths in $\Lambda_i^{t,l}$. The pseudo code is shown in Alg. 8 in Appx. B.1.

Given a node at level $t$, agents collectively search from the node to find the pivot time step (Line 13 in Alg. 1). The visit count is vital for search methods. In MARL, we use $\epsilon$-greedy [31] for agents to explore the environment and collect individual trajectories. The collected trajectories will be used to build the memory and train the policy. We apply annealing to $\epsilon$ (in Appx. E).

**Ranking the Pivot Timesteps.** With our two search schemes, we can search the pivot timesteps for each global trajectory $\tau = \{(s^t, \tilde{\boldsymbol{u}}^t, r^t, s^{t+1})\}_{t=0}^{T-1}$. We define the pivot timesteps $\kappa$ of each global trajectory $\tau$ as $\kappa = \{e_t\}_{t=0}^{T-1}$, $0 \leq e_t \leq t$, where $e_t$ indicates the pivot timestep of $t$ when $r_t$ is the consequence of actions committed before timestep $t$. We first get $e_t$ by aggregating all the searching outcomes (Line 13 in Alg. 1). Then, each agent gets $\kappa_i = \{e_t^i\}_{t=0}^{T-1}$. In order to subserve the inter-agent credit assignment [13, 41], $\kappa$ can be collectively calculated via proximity:

$$e_t = \min_{e_t^i} \left[ t - e_t^1, \cdots, t - e_t^N \right], \ i \in \{1, \cdots, N\} \tag{1}$$

The pseudo code is shown in Alg. 1. For each sampled global trajectory $\tau$, we extract $\tau_i$ for each agent in Line 7; then we get $e_t$ for each agent and aggregate $\kappa$ in line 14 and line 15, respectively.

## 5 Reward Redistribution for Off-Beat Multi-Agent Reinforcement Learning

Searching in LeGEM leverages the collective intelligence [25, 15] in OBMAS. We utilize TD learning to train MARL policies. The TD error is the difference between the TD target and the prediction. TD targets can be estimated with $n$-step target, TD($\lambda$) and other techniques [12, 56]. Unfortunately, current $n$-step target and TD($\lambda$) methods are far from accurate estimating TD targets. They even incur underestimation with off-beat trajectories. In essence, to train MARL policies in OBMAS, one should accurately estimate the TD target where the reward plays the key role [46, 70]. We resolve the aforementioned conundrum by redistributing rewards to their pivot timesteps. The key idea is that we can pull the outcome of one joint off-beat action back to the timestep when it was committed to the environment, which can dramatically enhance learning despite the long-term reward delays incurred by off-beat actions. We utilize $e_t$ to update the reward of the transit $(s^{e_t}, \tilde{\boldsymbol{u}}^{e_t}, r^{e_t}, s^{e_t+1})$:

$$\hat{r}^{e_t} = \mathbb{1}(e_t \geq t) \cdot r^{e_t} + \mathbb{1}(e_t < t) \cdot r_t, \tag{2}$$

where $\mathbb{1}(\cdot)$ is the indicator function. Such update rule is conducted iteratively from $t = 0$ to $t = T - 1$. $\beta$ is a very small positive hyperparameter. To stabilize learning and circumvent the overestimation of the TD target, $r_t$ is also updated after Eqn. 2 via $r_t = (1 - \mathbb{1}(e_t < t) \cdot (1 - \beta)) \cdot r_t$. It also avoids aggregated biased/wrong estimation of TD target being back propagated in Bellman Equation. Formally, we define the reward redistribution operator as $\Pi_\Phi$, *i.e.*, $e_t = \Pi_\Phi \rho(r^t, \boldsymbol{s}, \tilde{\boldsymbol{u}})$, and then define the Off-Beat Bellman operator $\Gamma$:

$$(\Gamma Q^{\text{tot}})(\boldsymbol{s}, \tilde{\boldsymbol{u}}) := \mathbb{E}[\Pi_\Phi R(\boldsymbol{s}, \tilde{\boldsymbol{u}}, \boldsymbol{m}) + \gamma \max_{\tilde{\boldsymbol{u}}'} Q^{\text{tot}}(\boldsymbol{s}', \tilde{\boldsymbol{u}}')] \tag{3}$$

With the Off-Beat Bellman operator $\Gamma$, we propose its contraction property.

**Proposition 1.** $\Gamma : \mathcal{Q} \mapsto \mathcal{Q}$ *is a $\gamma$-contraction.*

Therefore, we can utilize $\hat{r}_{e_t}$ for *centralized training* in TD-learning:

$$\mathcal{L}^{\text{TD}}(\theta) := \mathbb{E}_{\mathcal{D}' \sim \mathcal{D}}[(\hat{y}_{e_t}^{\text{tot}} - Q_\theta^{\text{tot}}(\boldsymbol{s}^{e_t}, \tilde{\boldsymbol{u}}^{e_t}))^2], \text{ where } \hat{y}_{e_t}^{\text{tot}} = \hat{r}^{e_t} + \gamma \max_{\tilde{\boldsymbol{u}}'} Q_{\hat{\theta}}^{\text{tot}}(\boldsymbol{s}^{e_t+1}, \tilde{\boldsymbol{u}}'). \tag{4}$$

Our method can be easily incorporated into any model-free MARL method for OBMAS. We present the pseudo code of incorporating our method into model-free MARL methods in Alg. 9, Appx. E. We also provide a pictorial view of our framework in Fig. 3 to show the whole pipeline.

# 6 Experiments

We perform experiments on various multi-agent scenarios with off-beat actions. We introduce off-beat actions in Stag-Hunter Game, Quarry Game, Afforestation Game and StarCraft II microman-agement tasks [44] and use them as testbeds in our experiments. We aim to answer the following questions: **Q1:** *Can our LeGEM improve the multi-agent coordination of many MARL methods in OBMAS?* **Q2:** *Can our LeGEM outperform previous parameterized episodic memory (EM) for MARL?* **Q3:** *Can bootstrapping method of RL help?* **Q4:** *Can our LeGEM outperform the multi-agent exploration and multi-agent risk-sensitive (Ex-Risk) methods?*

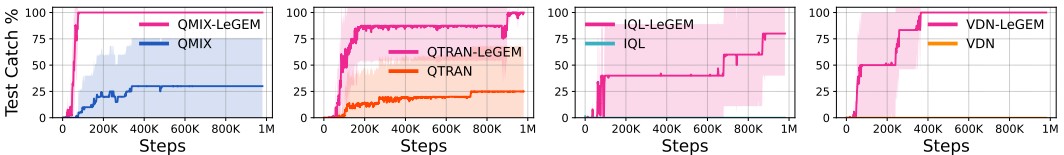

Figure 4: The test catch rate of the stag on the Stag-Hunter Game with off-beat actions.

## 6.1 Experiment Setup

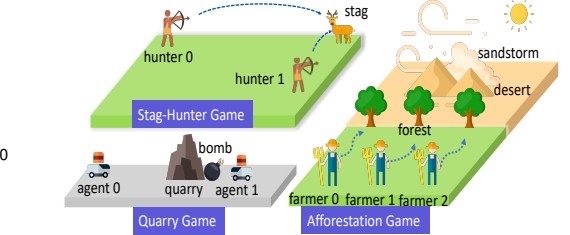

Figure 5: Stag-Hunter Game, Quarry Game and Afforesta-tion Game. More information can be found in Appx. C.

| Categories | Methods |
|---|---|
| MARL (**Q1**) | QMIX [41], VDN [49] IQL [53], QTRAN [47] QPLEX [60] |
| EM (**Q2**) | EMC [71] |
| Bootstrap (**Q3**) | N-step & $\lambda$-Return [51] |
| Ex-Risk (**Q4**) | MAVEN [28], EMC [71] RMIX [38] |

Table 1: Baseline algorithms.

**Baselines and scenarios.** We list all baselines in table 1, including the corresponding research questions to be answered. We implement our method on PyMARL [44] and use 10 random seeds to train each method on all environments. We do not use macro-action methods [67, 68] as the baseline because it is hard to make a fair comparison between macro-actions methods and our method. As discussed in Sec. 1, macro-actions rely on manually designed macro-actions, *i.e.*, designing the macro-actions by utilizing the simulator settings and domain knowledge, which is different from learning options [52, 3]. Designing macro-actions is not feasible in scenarios where domain knowledge and simulator settings are unknown, such as the OBMAS scenarios. In OBMAS, the agent has no idea of the durations of other agents' actions, which is challenging for designing macro-actions. We conduct experiments on Stag-Hunter Game, Quarry Game, Afforestation Game (Fig. 5) and StarCraft II micromanagement tasks [44] where off-beat action are introduced.

**Training settings.** We use opensourced code of baselines publicly by the corresponding authors on Github in all experiments. We resort to mean-std values as our performance evaluation measurement in all figures where the bold line and the shaded area indicate the mean value and one standard deviation of the episode return, respectively. Readers can refer to Appx. C, D, E and F for more information on our environment, baselines, training method, training platform and empirical results.

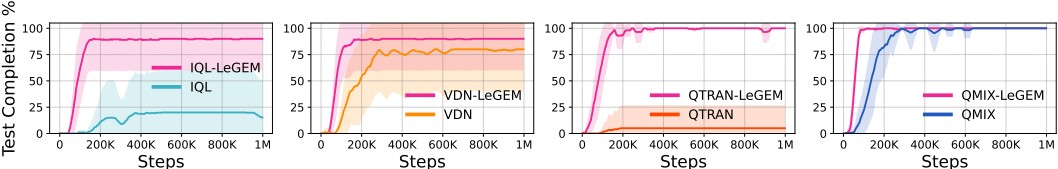

Figure 6: The test task completion rate of the Quarry Game with off-beat actions.

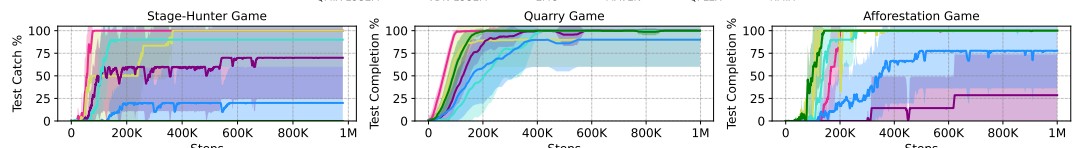

Figure 7: Performance of MARL methods

## 6.2 Experiment Results

**The Effectiveness of LeGEM.** We answer **Q1**. With LeGEM, MARL methods get enhanced performance as shown in Fig. 4. Without LeGEM, all methods perform poorly in Stag-Hunter Game; IQL and VDN's final final results are even 0. By incorporating LeGEM, all of them can get converged performance and improved sample efficiency. We are also interested in finding if LeGEM could reinforce the performance of simple methods. As depicted in Fig. 7, with LeGEM, both VDN and QMIX outperforms QPLEX, which is a state-of-the-art MARL method armed with various advanced techniques, including attention network [57], dueling network [64] and advantage function.

**Performance of Episodic Memory method.** We answer **Q2** by presenting the performance curves of EMC in Fig. 7. EMC is an episodic memory MARL method with curiosity-driven exploration. It utilizes the episodic memory from RL [74, 17].With LeGEM, QMIX outperforms EMC. EMC even fails to converge in Stag-Hunter Game.

Table 2: Results (mean and std) of $n$-step return (left) and TD($\lambda$) (right) on Stag-Hunter Game.

| $n$ | 1 | 5 | 10 | 15 | $\lambda$ | 0.8 | 0.9 | 0.99 | 1 |
|------|------|------|------|------|------|------|------|------|------|
| QMIX | $60.0 \pm 40\%$ | $0 \pm 0$ | $0 \pm 0$ | $0 \pm 0$ | QMIX | $100 \pm 0\%$ | $100 \pm 0\%$ | $89 \pm 10\%$ | $61 \pm 37\%$ |
| VDN | $0 \pm 0$ | $0 \pm 0$ | $0 \pm 0$ | $0 \pm 0$ | VDN | $0 \pm 0$ | $0 \pm 0$ | $0 \pm 0$ | $0 \pm 0$ |

**Performance of $n$-step return and TD($\lambda$) methods.** To answer **Q3**, we use $n$-step return and TD($\lambda$) to estimate the TD-target. As shown in Table. 2, with $n$-step return, both QMIX and VDN fail to learn good policies even with $n = 15$. Surprisingly, with TD($\lambda$), QMIX can achieve good performance with $\lambda \in \{0.8, 0.9, 0.99, 1\}$. However, we cannot find such outcome on VDN and there is no guarantee of good results on using TD($\lambda$).

**Performance of Multi-Agent Exploration and Risk-Sensitive MARL methods.** We also provide results of exploration methods for MARL and risk-sensitive MARL method to answer **Q4**. MAVEN utilizes mutual information to learn latent space for exploration and RMIX aims to learning risk-sensitive policies for MARL. In Fig. 7, RMIX even fails to learn. Mainly because the potential loss of reward is displaced by off-beat actions. Overall, MAVEN is stabler than EMC and RMIX. QMIX-LeGEM is stable in all scenarios and outperforms MAVEN. With LeGEM, even simple method such VDN can perform well and out-

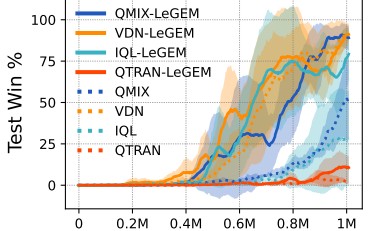

Figure 8: The performance of MARL methods on 2m_vs_1z.

performs many MARL methods with complex and advanced components. Indeed, exploration in OBMAS is beneficial for multi-agent learning. However, the key challenge of temporal credit assignment can not be easily addressed merely with exploration.

**SMAC.** We also conduct experiments on SMAC [44]. We train MARL methods and our method on 2m_vs_1z where are two agents combating with one opponent. To overcome the issue of

high dimension continuous state space, We utilize simhash [9] to calculate the hash value of the key. We only select the attack action and set the action duration with 9. As illustrated in Fig. 8, incorporated with our novel episodic memory, QMIX, IQL and VDN illustrate enhanced performance, demonstrating the superiority of our method on complex multi-agent scenarios.

## 7   Related Works

**Action Delay in RL.** Conventionally, the execution of actions in RL is instantaneous and the execution duration is neglected. Katsikopoulos et al. [20] propose the Delayed MDP where actions have delays and Walsh et al [59] propose a model-based method for the Delayed MDP. To optimize the delayed MDP, many RL approaches [59, 39, 66, 69] augment the state space with the queuing actions to be executed into the environment. However, this state-augmentation trick is intractable [11]. Chen et al. [10] extend the delayed MDP [39] and propose a Delayed Markov Game. However, the state-augmentation treatment is confined to short delays and neglects the off-beat actions in multi-agent scenarios. Recently, Bouteiller et al. [6] apply replay buffer correction method. However, the delayed timestep is privileged information. It is not available for agents in many scenarios. Simply applying this single-agent trajectory correction in MARL cannot attain satisfactory performance due to off-beat actions; devising inter-agent trajectory correction methods for OBMAS is non-trivial.

**Credit Assignment in RL.** Credit assignment [50, 52] tackles long-horizon sequential decision-making problem by distributing the contribution of each single step over the temporal interval. TD learning [51] is the most established credit assignment method, which is the basis of many RL methods. RUDDER [2] redistributes the episodic return to key timesteps in the episode [14, 42, 40]. Klissarov et al. [22] propose a reward propogation method via graph convolutional neural network [21]. Another line of works utilize episodic memory (EM) [37, 5, 73, 27, 74] to recall key events and aggregate information of the past for decision-making or learning. However, simply applying EM of RL to MARL cannot perform well in OBMAS due to the non-stationarity and the displaced rewards.

**Multi-Agent RL.** Many MARL methods focus on factorizing the global Q value to train agents' policies via CTDE [13, 49, 41, 47, 60, 63, 35]. However, these existing works assume actions are executed synchronously. Messias et al. [30] propose an event-driven, asynchronous formulation of the multi-agent POMDP. However, the assumption of free communication [61] is limited and the asynchronous execution [34] in the paper is confined to the design of events and did not propose methods on solving challenging credit assignment issue in OBMAS. Recently, Amato et al. [1] and Xiao et al. [67, 68] propose macro-action methods, which are similar to hierarchical methods. Macro-actions are manually designed via abstracting primitive actions. However, macro-action methods mainly focus on macro-action selection during multi-timestep decision-making and assume the environment can use manually pre-defined methods for state transition. Unfortunately, the above works either focus on synchronous actions or defining specific asynchronous execution components with human knowledge. Learning coordination in OBMAS remains a challenge.

## 8   Conclusion

In this paper, we investigate model-free MARL with off-beat actions. To address challenges in OBMAS, we first propose Off-Beat Dec-POMDP. Then, we propose a new class of episodic memory, LeGEM, for model-free MARL algorithms. LeGEM addresses the challenging temporal credit assignment problem raised by off-beat actions in TD-learning via the novel reward redistribution scheme. We evaluate our method on various OBMAS scenarios. Empirical results show that our method significantly boosts the multi-agent coordination and achieves leading performance as well as improved sample efficiency.

**Limitations and Future Work.** Searching from a graph-structured episodic memory takes much overhead in LeGEM. Scaling up LeGEM to complex OBMAS is our future direction. Recently, there is a growing interest in model-based planing [45]. Leveraging LeGEM for model-based planning is also our future work. Our paper focuses on Dec-POMDP-based MARL methods. We leave it to future work for investigating off-beat actions in frameworks like Markov Game [26] and MMDP [7]. We are also interested in finding the merit of our method in real-world problem in our future work, such as scheduling [29] with off-beat settings.

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
