# OpenReview forum: "Off-Beat Multi-Agent Reinforcement Learning"
_NeurIPS.cc/2022/Conference — NeurIPS 2022 Submitted_

### Official Review · Reviewer_dJd4 · 2022-06-18

**Rating:** 5
**Confidence:** 3
**Soundness:** 2 fair
**Presentation:** 1 poor
**Contribution:** 2 fair

**Summary:**

This paper looks at the problem of coordination games where actions may have a delayed impact on the environment, which are referred to as "off-beat actions". To appropriately credit these off-beat actions they proposed a graph-based memory structure (LeGEM). It works by building a count-based graph of possible histories and searches for differences in graphs to assign future rewards to off-beat actions. They demonstrate their algorithm's ability to encourage coordination on three analytic games and the StarCraft II micromanagement task suite. Their results suggest LeGEM leads to a higher probability of successful coordination.

**Questions:**

1. Could the authors explain how modelling the actions as having a delayed effect on the transition dynamics is usefully different here than just considering that the reward might be delayed?

2. It seems like the multi-agent aspect of this problem is simply that there are decentralized unit-actions. Why did the authors not consider comparing this work to single-agent off-beat algorithms? It seems that they might be applied directly or with small modifications and provide reasonable baselines.

3. I am hoping the authors might better justify their choice of model for off-beat actions. Initially the exposition introduces them like a sub-system within the environment dynamics, but they're modelled as a process that occurs prior to transition. Moreover, the off-beat calculation is ego-tistically considering only a single agent. However, this paper focus is on games, should off-beat actions not consider other-agents and their influence on delays?

4. Could the authors please clarify how their model "find[s] an off-beat reward in a trajectory" (L171). All formalisms seem to suggest that an agent receives a single scalar reward at each timestep, but it's not clear how this could be decomposed into rewards from multiple co-occurring off-beat action rewards?

5. Nit: the line spacing on page 6 is off.

6. L124-5: "there is no MARL method that can explicitly recall the past and identify key states that lead to future reward" This seems like a pretty large claim that I don't think is genuine. Tabular methods have explicitly built state-return mappings in the past.

7. Could the authors please comment on how this method compares to reward shaping. It seems like reward shaping methods might be comparable baselines as it could amount to modifying the credit assignment pattern for an agent.

8. Do the authors have any analysis or ablation studies on the different components of their algorithm? It would be interesting to see which search patterns / heuristic are useful and under what settings a practioneer may want to use them (or not).

**Limitations:**

The authors should include their limitation discussion to note that (a) this method only works for strictly cooperative games, and (b) methods where all information can be perfectly shared across all agents.

**Strengths And Weaknesses:**

**Strengths**
- Addresses, indirectly, a challenging problem of ascribing delayed-credit to actions.

**Weaknesses**
- As a non-expert on "off beat" systems I found the introductory exposition challenging to read. An "off beat" action is not fully defined until page 3. I would suggest the authors provide motivating examples and describe the particular problem of "off beat" actions that they will study much earlier in the paper.
- There are many claims that are not appropriately supported. Non-exhaustive list of examples: L38-44 contains many "challenges' of OBMAS that is neither cited nor proven why this is a quality and problem unique to OBMAS; L52-53, it's not super clear at this point what is meant by "delayed timestep of the actions" or why this might be privileged information and why this is meaningful; L236 Proposition 1 is not proved, you could imagine an adversarial reward redistribution that prevents a contraction.
- The experiments are focused around "outperforming" various MARL methods, but not on understanding the method and problem proposed.

---

> ### Author Response · Authors · 2022-07-30
> **Response to Reviewer dJd4 1/2**
>
> Dear reviewer, we thank your valuable comments and questions on our paper. We summarize your questions and present our responses below. More questions and discussions on our paper are welcomed!
>
> ---
>
> **Q1: An "off beat" action is not fully defined until page 3. I would suggest the authors provide motivating examples and describe the particular problem of "off beat" actions that they will study much earlier in the paper.**
>
> **A1:** In the abstract, we introduce off-beat action in Lines 1-5.
>
> > We investigate model-free multi-agent reinforcement learning (MARL) in environments where off-beat actions are prevalent, i.e., all actions have pre-set execution durations. During execution durations, the environment changes are influenced by, but not synchronised with, action execution. Such a setting is ubiquitous in many real-world problems.
>
> We then introduce off-beat actions in Lines 28-31.
>
> > One key challenge is the off-beat actions, i.e., all actions have pre-set execution durations and during the execution durations, the environment changes are influenced by, but not synchronised with, action execution (an illustrative scenario is shown in Fig. 1).
>
> We also provide an example in Fig. 1.
>
> ---
>
> **Q2: L38-44 contains many "challenges' of OBMAS that is neither cited nor proven why this is a quality and problem unique to OBMAS;**
>
> **A2:** With off-beat actions, challenge (1) exposes challenges for multi-agent coordination, which has been proved in Fig. 1 and our experiments. We also discussed related works on cooperative MARL in Sec. 7. In Lines 38-40, we will cite these related papers.
>
> For challenge (2), in previous RL works, the action duration value is accessible, making it much easier to solve. We recommend the reviewer read the “Action Delay in RL” paragraph in Sec. 7. However, the action duration value may not always be available in MARL. The exact action duration value is not accessible to agents in our setting. We will also cite these related papers.
>
> For challenge (3), in TD learning, the rewards are displaced by the off-beat actions, which harms the MARL training. We cited some papers in Lines 42-43.
>
> ---
>
> **Q3: L52-53, it's not super clear at this point what is meant by "delayed timestep of the actions" or why this might be privileged information and why this is meaningful;**
>
> **A3:** The “delayed timestep of the actions” means the exact duration of action. For example, one action may have a duration of 5 (timesteps). In
>
> ---
>
> **Q4: L236 Proposition 1 is not proved, you could imagine an adversarial reward redistribution that prevents a contraction.**
>
> **A4:** We present the proof in Sec. A in the Appendix. It is available in the supplementary folder. If we are not wrong, we guess you are talking about the adversarial attack on reward redistribution. The adversarial setting is not the focus of our paper and we could consider it in future works.
>
> ---
>
> **Q5: The experiments are focused around "outperforming" various MARL methods, but not on understanding the method and problem proposed.**
>
> **A5:** We propose off-beat actions, off-beat Dec-POMDP and off-beat MARL. Our novel method aims to solve the problem raised by off-beat actions. We also explain our method in the first paragraph, Sec. 4.1 and the paragraph 1-4, Sec. 4.2.
>
>
> ---
>
> **Q6: Could the authors explain how modelling the actions as having a delayed effect on the transition dynamics is usefully different here than just considering that the reward might be delayed?**
>
> **A6:** We think the delayed rewards are the outcome of off-beat actions. Agents interact with the environment by committing actions. The outcome is the displaced reward. In our formulation (see Definitions 1 and 2; Remarks 1-4 in Sec. 3.), the rewards are delayed or displaced with off-beat actions. So, in our method, we solved the problem via reward redistribution.
>
> ---
>
> **Q7: It seems like the multi-agent aspect of this problem is simply that there are decentralized unit-actions. Why did the authors not consider comparing this work to single-agent off-beat algorithms?**
>
> **A7:** Our paper considers the problem of cooperative multi-agent RL with off-beat actions. If we are not wrong, we assume the “single-agent off-beat algorithms” you mentioned are RL algorithms that tackle reward delays. We tried experiments by adding the delay (action duration value) into the inputs. However, it cannot work and the action duration value is unavailable in our setting. Besides, “single-agent off-beat algorithms” were not designed for cooperative MARL; adapting these methods into the MARL setting is non-trivial. We will put the results of RUDDER in the paper during the rebuttal phase.

---

> > ### Author Response · Authors · 2022-07-30
> > **Response to Reviewer dJd4 2/2**
> >
> > **Q8: I am hoping the authors might better justify their choice of model for off-beat actions. Initially the exposition introduces them like a sub-system within the environment dynamics, but they're modelled as a process that occurs prior to transition. Moreover, the off-beat calculation is ego-tistically considering only a single agent. However, this paper focus is on games, should off-beat actions not consider other-agents and their influence on delays?**
> >
> > **A8:** Yes, but the environment dynamics control how off-beat actions work in the environment, such as controlling the flying arrow hitting the prey. Off-beat actions have durations. Once one off-beat action is committed to the environment, it takes some time to complete. Consider the case where a hunter shoots an arrow and takes some time to fly and hit the target prey. The flying arrow may hit other prey on the way to the target prey. So, this case is not controlled by the agent but by the environment.
> >
> > ---
> >
> > **Q9: Could the authors please clarify how their model "find[s] an off-beat reward in a trajectory" (L171). All formalisms seem to suggest that an agent receives a single scalar reward at each timestep, but it's not clear how this could be decomposed into rewards from multiple co-occurring off-beat action rewards?**
> >
> > **A9:** It is worth noting that all agents globally share rewards in Dec-POMDP and Off-beat Dec-POMDP settings. Once an off-beat reward in a trajectory is found, LeGEM will search from the node at that time step in the graph and return the pivot time step without decomposition. We use the ranking method to get the pivot time step since each agent can use the reward to search the pivot time in its graphs. Each agent can return the searched pivot time steps, and we should decide the best time step. Then, the reward will be redistributed. The mixing networks in all baseline MARL methods conduct inter-agent credit assignment.
> >
> > ---
> >
> > **Q10: L124-5: "there is no MARL method that can explicitly recall the past and identify key states that lead to future reward" This seems like a pretty large claim that I don't think is genuine. Tabular methods have explicitly built state-return mappings in the past.**
> >
> > **A10:** After reading over 150 papers, we found that “there is no MARL method that can explicitly recall the past and identify key states that lead to future reward”. Could you please provide these related papers? It would be great for us to cite them and change our wording.
> >
> > ---
> >
> > **Q11: Could the authors please comment on how this method compares to reward shaping. It seems like reward shaping methods might be comparable baselines as it could amount to modifying the credit assignment pattern for an agent.**
> >
> > **A11:** We will present the results of RUDDER [1] during the rebuttal phase. RUDDER is one of the most representative reward shaping methods.
> >
> > ---
> >
> > **Q12: Do the authors have any analysis or ablation studies on the different components of their algorithm?**
> >
> > **A12:** Yes. Our main component is LeGEM. Analysis of our method is in Sec. 4.2. We also provide results without LeGEM in Sec. 6.2.  Besides that, we tried two other ranking methods:
> >
> > (1) We redistribute the reward to the searched pivot time step t, which is the farthest time step from the time step of the reward. At time step t, an off-beat action was committed to the environment. However, the result is not as good as the one presented in our paper. The reasons are (i) the action taken at time step t is not the key action to the reward. In OBMAS, for example, in the scenario in Fig. 1 in our paper, action SHOOT is taken by agent one at time step 5 is the key action to the reward at time step 9; (ii) In Dec-POMDP MARL methods, we use RNN in the policy network to mitigate the issue raised by partial observation. The RNN can backpropagate the redistributed reward at the pivot timestep to the time steps before it. Besides that, the Bellman update can also backpropagate the redistributed reward to Q values of state-action pairs before the pivot timestep.
> >
> > (2) We redistribute reward at time step t (with LeGEM, the pivot time step is t’) to all time steps where off-beat actions were taken. This scheme did not perform well either. The main reason is that the redistributed reward to time steps before the time step t’ can overweigh the corresponding Q values.
> >
> > ---
> >
> > **References:**
> >
> > [1] J. A. Arjona-Medina, M. Gillhofer, M. Widrich, T. Unterthiner, J. Brandstetter, and S. Hochreiter. RUDDER: Return decomposition for delayed rewards. In Advances in Neural Information Processing Systems, volume 32, 2019.

---

> > > ### Comment · Reviewer_dJd4 · 2022-08-08
> > > **Response**
> > >
> > > Thank you for taking the time to address my points.
> > >
> > > As it currently stands I am still not persuaded that off-beat modelling is a meaningful model choice that offers advantages. Despite this, my fellow reviewers do not share this same concern so I will raise my score, but I urge the authors to strongly consider each modeling choice and what advantages/disadvantages it affords you (e.g., what makes it meaningfully different from another similar choice).

---

> > > > ### Author Response · Authors · 2022-08-09
> > > > **Thank You Reviewer dJd4**
> > > >
> > > > Dear Reviewer dJd4,
> > > >
> > > > We deeply appreciate your opinion about off-beat modeling in our paper. We thank you for raising the score.
> > > >
> > > > We are grateful to have fruitful discussions with you! We introduced off-beat actions in Dec-POMDP, a very popular MARL modelling method. Many cooperative MARL methods, including COMA [1], QMIX [2] and QPLEX [3], apply this model to model cooperative multi-agent problems. We presented our solution in our paper to address issues caused by off-beat actions. We leave it to future work (in Sec.8, page 9) to investigate off-beat actions in frameworks like Markov Game and MMDP because we find the key problem of off-beat actions in these modelling frameworks is similar, i.e., the credit assignment issue  (discussed in Sec. 1) caused by the displaced rewards due to off-beat actions.
> > > >
> > > > Thank you again for your feedback and your effort in reviewing our paper!
> > > >
> > > > Sincerely yours,
> > > >
> > > > Paper5095 Authors
> > > >
> > > > ---
> > > >
> > > > **References**
> > > >
> > > > [1] Foerster, Jakob, et al. "Counterfactual multi-agent policy gradients." Proceedings of the AAAI conference on artificial intelligence. Vol. 32. No. 1. 2018.
> > > >
> > > > [2] Rashid, Tabish, et al. "Qmix: Monotonic value function factorisation for deep multi-agent reinforcement learning." International conference on machine learning. PMLR, 2018.
> > > >
> > > > [3] Wang, Jianhao, et al. "QPLEX: Duplex Dueling Multi-Agent Q-Learning." International Conference on Learning Representations. 2020.

---

> ### Author Response · Authors · 2022-08-04
> **Dear Reviewer dJd4, did our responses address your concerns?**
>
> Dear Reviewer dJd4,
>
> As the response system will be closed soon within one week. We thank you again for your valuable comments. We made responses to your questions. We hope our responses can address your questions. More questions on our paper are always welcomed! If there are no more questions, we would appreciate it if you could kindly raise the score.
>
> Sincerely yours,
>
> Paper5095 Authors

---

> ### Author Response · Authors · 2022-08-08
> **Dear Reviewer dJd4, did our responses clear up your confusion?**
>
> Dear Reviewer dJd4,
>
> We appreciate your valuable comments on our paper. We made responses to introduce our methods further to **non-expert readers** (updated in Appendix) and **address your concerns** (adding experimental results of RUDDER). The author-reviewer discussion period will end soon, and we look forward to hearing your valuable comments on our responses.
>
>
> Sincerely yours,
>
> Paper5095 Authors

---

### Official Review · Reviewer_Y4Go · 2022-07-06

**Rating:** 6
**Confidence:** 4
**Soundness:** 3 good
**Presentation:** 2 fair
**Contribution:** 3 good

**Summary:**

This work introduces a novel generalization of Dec-POMDPs that also takes different execution durations for actions into account. Furthermore, the authors propose a novel approach that uses graph algorithms to search for critical points in agent trajectories. These points are used to assign rewards to the corresponding action. In a series of multi-agent tasks, which are modified such that they exhibit off-beat actions, LeGEM enhanced algorithms are compared to baselines. The baseline algorithms span simple MARL algorithms to memory enhanced, more sophisticated approaches.

**Questions:**

- The experiments only involve a few agents. How is the performance influenced by adding more agents, e.g. up to 100's? It would be valuable to have one experiment that explores this.
- From the experiments you run, how much overhead is approximately introduced by the graph updates/search?

**Limitations:**

The authors admit that their approach introduces a large computational overhead and that scalability was not addressed in this paper.

**Strengths And Weaknesses:**

## Originality

The paper introduces off-beat Dec-POMDPs, a novel generalization of delayed MDPs and Dec-POMDPs. The approach uses levelled-graphs to represent the episodic memory. This apprears to be a novel combination of levelled-graphs and deep MARL methods.

*Minor Remark:* While you include [74] in the experiments section, it should also be cited in section 4.1 as another approach that uses graphs to structure EM.

**Strengths:**

- Novel application of established data structures/algorithms to deep RL

**Weaknesses:
-**

## Quality

The approach seems to be valid in general. The experimental evaluation is a major strength of this paper. First, the authors identify relevant questions to address. The experiment design is adequate as it demonstrates the efficacy of their approach along several dimensions, such as performance improvements of existing approaches and comparisons to alternative episodic memory approaches. Thus, the experiments undermine the claimed contributions.
It would have been interesting to see how much computational overhead (e.g. training time) is induced by the graph updates and the search. You may consider to include some comparisons in this direction to other EM approaches. Furthermore, investigating the performance of LeGEM in many-agent tasks would be important.

**Strengths:**

- Algorithms appear to be sound
- Well formulated questions to answer
- Strong experimental evaluation: well chosen baselines and experiments

**Weaknesses:**

- No experiment addressing computational overhead
- Experiments only involve a few ($\leq$ 3) agents.

## Clarity

In general, the paper is well written. However, the notation that is introduced is rather heavy (many indices, sub-indices, etc...) and uses many symbols. This is especially true for section 4 and the algorithm pseudo code. Also, $\Upsilon$ is not defined in the main paper, only in the appendix. Also, the training procedure (Fig. 3) could be addressed more clearly.
Browsing through the code helped when trying to understand the algorithms. However, it would have been nice to also include a file that runs an experiment or trains a policy. Why did you not include the training script?

**Strengths:**

- Generally well written paper.
- Code was provided.

**Weaknesses**:

- The amount of symbols and indexed symbols makes it somewhat hard to read.
- No training script/executable included.

## Significance

The problem addressed in the paper, off-beat actions, is novel and most probably important for real world applications of RL. Furthermore, the proposed approach could be a stepping stone for other works exploring this new problem setting. Also the evaluation of other algorithms on this kind of problem is valuable.
The performance was demonstrated on simple tasks. The results could be more convincing if you would include an experiment with more agents (e.g. 20).

**Strengths:**

- Significant problem with plausible real-world connection

**Weaknesses:**
- The tasks that were solved are rather simplistic.

---

> ### Author Response · Authors · 2022-07-30
> **Response to Reviewer Y4Go**
>
> Dear reviewer, we thank your valuable comments and insightful suggestions on our paper. We summarize your questions and present our responses below.
>
> ---
>
> **Q1: The experiments only involve a few agents. How is the performance influenced by adding more agents, e.g. up to 100's? It would be valuable to have one experiment that explores this.**
>
> **A1:** Scaling up MARL methods to a large number of agents is non-trivial (see some MARL related works in Sec. 7). We leave it to future work to invest the problem of learning cooperative MARL in OBMAS with a large number of agents.
>
> ---
>
> **Q2: How much overhead is approximately introduced by the graph updates/search?**
>
> **A2:** The search cost in the graph is proportional to the length of the graph. In Stag-Hunter Game, Quarry Game and Afforestation Game, it will cost 100% extra time in training. SMAC scenarios will cost ~300% extra time in training. The search time is also highly dependent on the implementation. With Python, it can be large than that with C++ implementation.
>
> ---
>
> **Q3: The amount of symbols and indexed symbols makes it somewhat hard to read**
>
> **A3:** To better understand the paper, we also provide a list of symbols in Sec. B.3 in the Appendix. We will highlight it in our paper.
>
> ---
>
> **Q4: No training script/executable included.**
>
> **A4:** We will put the training scripts in the revised version. We get the code of each baseline from the corresponding GitHub repository. The code of QMIX, VDN, IQL and QTRAN are from PyMARL’s Github repository: https://github.com/oxwhirl/pymarl, which has been widely used in the MARL community.
>
> ---
>
> **Q5: Minor Remark: While you include [74] in the experiments section, it should also be cited in section 4.1 as another approach that uses graphs to structure EM**
>
> **A5:** We will cite it in the revised version.

---

> > ### Comment · Reviewer_Y4Go · 2022-08-03
> > **Response to the authors**
> >
> > Thank you for the answers.
> >
> > >A1: Scaling up MARL methods to a large number of agents is non-trivial (see some MARL related works in Sec. 7). We leave it to future work to invest the problem of learning cooperative MARL in OBMAS with a large number of agents.
> >
> > Ok, maybe not 100's but it would be definitely interesting to see the scaling behavior of your approach, given that you mention a 100-300% time overhead.
> >
> > >A2: The search cost in the graph is proportional to the length of the graph. In Stag-Hunter Game, Quarry Game and Afforestation Game, it will cost 100% extra time in training. SMAC scenarios will cost ~300% extra time in training. The search time is also highly dependent on the implementation. With Python, it can be less than that with C++ implementation.
> >
> > Thank you for the information. You should include this in the paper. Maybe a complexity analysis w.r.t different factors (such as number of agents, number of timesteps, etc.) would also be valuable if it is doable within the time frame of he rebuttal phase.
> >
> > >A3: To better understand the paper, we also provide a list of symbols in Sec. B.3 in the Appendix. We will highlight it in our paper.
> >
> > I consulted this table during the review. However, the paper introduces a lot of different symbols and algorithms which are difficult to keep in mind when going through the pseudocode (especially due to the high number of indices). Maybe you can find a way to ease the notation.

---

> > > ### Author Response · Authors · 2022-08-09
> > > **Thank You for Your Suggestions**
> > >
> > > Dear Reviewer Y4Go,
> > >
> > > We thank you again for recognising the contribution of our work to the multi-agent system and multi-agent RL community. We appreciate your time and hard work in providing insightful suggestions for our paper.
> > >
> > > Sincerely yours,
> > >
> > > Paper5095 Authors

---

### Official Review · Reviewer_eRuo · 2022-07-07

**Rating:** 5
**Confidence:** 4
**Soundness:** 3 good
**Presentation:** 3 good
**Contribution:** 3 good

**Summary:**

This paper addresses the issue in off-beat multi-agent reinforcement learning, where the actions have pre-set execution durations. During execution, the environment changes are influenced by, but not synchronized with, action execution. This off-beat issues exit in scenarios, such as traffic lights and video games. To mitigate this issue, the authors propose an episode memory module, i.e., Levelled Graph Episodic Memory (LeGEM) method. The authors propose a search method to search the pivot timestep and utilize a proximal ranking method to estimate the pivot timestep, which invokes the future reward. The authors evaluate the proposed method in different scenarios, including Stag-Hunter Game, Quarry Game, Afforestation Game, and StarCraft II micromanagement tasks. The empirical experimental results show that LeGEM can boost the performance of existing RL such as IQL, VDN, QTRAN, and QMIX.

**Questions:**

* How does LeGEM compare to other credit assignment methods, such as RUDDER?
* How does the training time of IQL, VDN, QTRAN, and QMIX compare to the training time of IQL-LeGEM, VDN-LeGEM, QTRAN-LeGEM, and QMIX-LeGEM?
* Does LeGEM also improves the performance in the general environments without the off-beat issue?

**Limitations:**

* The proposed method seems limited to the off-beats scenarios. It would be better if the proposed method improves the performance in the general environments.
* LeGEM introduces additional computational overhead. It might be difficult to utilize LeGEM in large scale settings.

**Strengths And Weaknesses:**

The strengths of the paper:

* The paper is well motivated and the problem is well defined.
* The proposed memory module shares connections with the human learning process and shows performance improvement in the experiments.
* The paper is well structured and well written. The figures in the paper help the reader to understand the idea.

The weaknesses of the paper:

* The baselines used in the experiments are relatively “weak”, including IQL, VDN, QTRAN, and QMIX, in the sense that they are not specially designed to solve the off-beat problem. Stronger baselines should be used. For example, the methods, which tackle the credit assignment problem should be compared, such as RUDDER [Arjona-Medina et al, 2019].
* The LeGEM module includes the search process. I have concerns about the training time.
* The off-beats problem exits but not very general.

---

> ### Author Response · Authors · 2022-07-30
> **Response to Reviewer eRuo**
>
> Dear reviewer, we thank your insightful questions and the valuable review. We summarize your questions and present our responses below.
>
>
> ---
>
> **Q1: IQL, VDN, QTRAN, and QMIX are not specially designed to solve the off-beat problem. Stronger baselines should be used.**
>
> **A1:** There are few methods proposed for solving off-beat actions. Our paper aims to learn cooperative MARL in off-beat multi-agent scenarios (OBMAS). So, we use some representative cooperative MARL methods to evaluate if our method can improve the performance of cooperative MARL methods in OBMAS.
>
> ---
>
> **Q2: The off-beats problem exits but not very general.**
>
> **A2:** If we are not wrong, we assume “exits” should be “exists” in your question. We believe Off-beat actions are general in real-world problems. Such as scheduling problems where each agent can have durations, making it an off-beat problem. We can also find off-beat actions in logistics, manufacturing, network flow control and grid control.
>
> ---
>
> **Q3: How does the training time of IQL, VDN, QTRAN, and QMIX compare to the training time of IQL-LeGEM, VDN-LeGEM, QTRAN-LeGEM, and QMIX-LeGEM?**
>
> **A3:** In the experiments, LeGEM is used in TD-learning. With LeGEM, it will cost 100% more time compared with training MARL methods without LeGEM. Despite the time cost, we can gain better performance with less samples.
>
> ---
>
> **Q4: Does LeGEM also improves the performance in the general environments without the off-beat issue?**
>
> **A4:** The reward will not be redistributed if there are no off-beat actions, as shown in Alg. 2 in the paper. So, in general, it will not help to improve the performance in the environment without off-beat actions.
>
> ---
>
> **Q5: LeGEM introduces additional computational overhead. It might be difficult to utilize LeGEM in large scale settings.**
>
> **A5:** Despite the time cost in searching, we believe the problem of off-beat action is a vital problem both in research and industries. The searching time costs depend on the implementation. With Python, the searching time is longer than that with C++ multi-threading. Besides that, solving large-scale problems is challenging even without off-beat actions for MARL.

---

> > ### Comment · Reviewer_eRuo · 2022-08-08
> > **To Authors:**
> >
> > Thank you for addressing my questions! After reading the rebuttal, I decide to keep my score.

---

> > > ### Author Response · Authors · 2022-08-09
> > > **Thank You for Your Response**
> > >
> > > Dear Reviewer eRuo,
> > >
> > > Thank you for your valuable responses.
> > >
> > > Sincerely yours,
> > >
> > > Paper5095 Authors

---

> ### Author Response · Authors · 2022-08-04
> **Dear Reviewer eRuo, did our responses clear up your confusion?**
>
> Dear Reviewer eRuo,
>
> We appreciate your effort in viewing our paper. We thank you again for your valuable comments. As the response system will be closed soon within one week. We hope our responses can address your questions. Did our responses clear up your confusion? More questions on our paper are always welcomed! If there are no more questions, we would appreciate it if you could kindly consider raising the score.
>
> Sincerely yours,
>
> Paper5095 Authors

---

### Official Review · Reviewer_CAwi · 2022-07-11

**Rating:** 5
**Confidence:** 3
**Soundness:** 2 fair
**Presentation:** 1 poor
**Contribution:** 2 fair

**Summary:**

This paper aims to address the delayed action problem in multi-agent reinforcement learning. The authors propose LeGEM, a episodic memory structure that improves the temporal credit assignment by searching for the pivot timestep and reward redistribution. The method is evaluated on four game domains.

**Questions:**

Please see the weakness 2-6 and comments on the experiments.

**Limitations:**

The authors address some of the limitations and I do not see the potential negative societal impact.

**Strengths And Weaknesses:**

**Strengths:**
This paper looks at an interesting and practical perspective of MARL, namely the delayed action. While it has been explored in the single-agent domain, there are fewer studies in the multi-agent domain.

**Weaknesses:**
I found it is overall hard to follow the paper:
1. The search scheme I part (Section 4.2) lacks explanation and intuition. There is little to no explanation of the notations in the algorithms., for example, $\Upsilon, \mathbf r^{l, i}, e^i_t[j]$, the meaning of -1 in Algorithms 2 etc. (and there are more examples like these in the paper). The missing information greatly impedes understanding.
2. Many new concepts come up without further mentioning or explanation, such as “subserve the inter-agent credit assignment”. What exactly is inter-agent credit assignment? How does it help?
3. Line 195: “the accuracy of Scheme I is over 90% in grid world scenarios.” What does “grid world” refer to? How is the accuracy defined?
4. I do not quite understand why Scheme II, a simplified version of Scheme I is able to deal with “more complex scenarios, whose trajectories are multiple-off-beat-reward trajectories”.
5. What is the role of $\beta$ in Equation (2)? How to interpret it?
6. Why does the off-beat bellman operator enjoy contraction property?

The experiment part is not convincing either:
* If the key is to identify the pivot timestep, shouldn’t there be some experiments to investigate the accuracy of that?
* The experiment environments are simple, with a maximum of three agents, short horizons, and small actions space. Could this be related to the scalability of the graph search method?

---

> ### Author Response · Authors · 2022-07-30
> **Response to Reviewer CAwi**
>
> Dear reviewer, we thank your time and effort in reviewing our paper. We summarize your questions and present our responses below. More discussions on our paper are welcomed!
>
> ---
>
> **General response:**
>
> Our paper investigates the action duration problem in MARL. The action duration problem has been investigated in RL. However, with action durations, actions in MARL become off-beat actions (i.e., asynchronous actions; see Remarks 1-4 in Sec.3 for more details). As many agents act in multi-agent scenarios, the challenges introduced by off-beat actions differ from action delays in RL (see paragraphs 3-4 in Sec. 1 for more information about the challenges).
>
> ---
>
> **Q1: The search scheme I part (Section 4.2) lacks explanation and intuition.**
>
> **A1:** As our search method relies on visit counts, we provided intuition in Lines 164-181 in our paper. We present an explanation: Given a reward at time step t, we are sure that the timestep of the action intrigued the reward is before t (c.f. Fact 1 in Lines 167-169). So, we can search from the level t upwards to find the pivot t. The node at the pivot time step is the key node, with many sub-paths starting from it. Naturally, we can find the pivot time step when the pattern of increasing visit count ends, and the corresponding level is the candidate pivot timestep. That is the Low-Up (LU) search. Such a pattern may not exist in some trajectories. Instead, we can search from level 0 downwards and find the pivot time step. In graph theory, finding the pivot time step can be considered as finding node co-occurrence in the graph.
>
>
> ---
>
>
> **Q2: There is little to no explanation of the notations in the algorithms.**
>
> **A2:** The notations of the algorithms are introduced in Sec. 4.1. We also provide lists of symbols in Sec. B.3, Table 3-5 in Appendix. The appendix is available in the supplementary folder.
>
> ---
>
>
> **Q3: What exactly is inter-agent credit assignment? How does it help?**
>
> **A3:** The inter-agent credit assignment decomposes the globally shared reward in Dec-POMDP for all agents as agents in the cooperative team may contribute differently to the global reward. We discussed in Line 42 that the reward displaced by the off-beat actions exposes challenges for TD learning and inter-agent credit assignment. There are many inter-agent credit assignment methods, such as QMIX [1], QTRAN [2] and QPLEX [3].
>
> ---
>
> **Q4: What does “grid world” refer to? How is the accuracy defined?**
>
> **A4:** The grid world is the simple scenario depicted in Fig. 1. For a simple scenario, we can find the ground-truth pivot time step and then calculate the accuracy of the search method.
>
> ---
>
> **Q5: Why is Scheme II able to deal with “more complex scenarios”?**
>
> **A5:** Because it has lower time complexity as analyzed in Sec. 4.2. Intuitively, Scheme II can be considered an attention method that finds the most appropriate time step for reward redistribution.
>
>
> ---
>
> **Q6: What is the role $\beta$  of  in Equation (2)? How to interpret it?**
>
> **A6:** $\beta$ controls the weight for updating the new $r_t$ after updating $\hat{r}^{e_t}$ with $r_t$. If $\beta$ is zero, it means after updating $\hat{r}^{e_t}$, $r_t$ will be set with the value of zero.
>
> ---
>
> **Q7: Why does the off-beat bellman operator enjoy contraction property?**
>
> **A7:** We proved it in Sec. A in Appendix. The appendix is available in the supplementary folder.
>
> ---
>
> **Q8: Shouldn’t there be some experiments to investigate the accuracy of that?**
>
> **A8:** We can calculate the accuracy if we can get the ground-truth pivot timestep. We ran the code, and the accuracy of our search method on Stag Hunt Game is 98%, on Afforestation Game is 98% and on Quarry Game is 99%. For complex (continues state/observation space) scenarios, such as SMAC, getting the ground-truth pivot time step is non-trivial due to the complexity of the simulator.
>
> ---
>
> **Q9: The experiment environments are simple. Could this be related to the scalability of the graph search method?**
>
> **A9:** Searching in a complex graph can be time-consuming, especially when the number of agents is large. We discussed this limitation in Lines 344-345, Sec. 8 in our paper.
>
> ---
>
> **References:**
>
> [1] T. Rashid, M. Samvelyan, C. Schroeder, G. Farquhar, J. Foerster, and S. Whiteson. QMIX: Monotonic value function factorisation for deep multi-agent reinforcement learning. In International Conference on Machine Learning, pages 4295–4304, 2018.
>
> [2] K. Son, D. Kim, W. J. Kang, D. E. Hostallero, and Y. Yi. QTRAN: Learning to factorize with transformation for cooperative multi-agent reinforcement learning. In International Conference on Machine Learning, pages 5887–5896, 2019
>
> [3] J. Wang, Z. Ren, T. Liu, Y. Yu, and C. Zhang. QPLEX: Duplex dueling multi-agent q-learning. arXiv preprint arXiv:2008.01062, 2020.

---

> > ### Comment · Reviewer_CAwi · 2022-08-06
> > **Response to Authors**
> >
> > Thanks for the clarification and revision. The authors have answered most of my questions. However, I still think the presentation could be improved by incorporating a more intuitive explanation of the graph search algorithm for non-experts. Nevertheless, I think it is solving an important problem. I am willing to raise my rating to "Borderline Acceptance".

---

> > > ### Author Response · Authors · 2022-08-08
> > > **Thank You Reviewer CAwi**
> > >
> > > Dear Reviewer CAwi,
> > >
> > > We are glad to hear that our responses addressed most of your questions. We thank you for raising the score and recognising the contribution of our paper to the multi-agent system and multi-agent RL community. In the newly updated paper, we incorporate an intuitive explanation of the graph search algorithm for non-experts readers. We hope the intuitive explanation can enhance your understanding of our method. More questions on our paper are always welcomed!
> > >
> > >
> > > Sincerely yours,
> > >
> > > Paper5095 Authors

---

> ### Author Response · Authors · 2022-08-04
> **Dear Reviewer CAwi, did our responses clear up your confusion?**
>
> Dear Reviewer CAwi,
>
> We thank you again for your valuable time and comments on our paper. We made responses to your questions. Did our responses clear up your confusion? More questions on our paper are always welcomed! As the response system will be closed soon within one week. If you have no more questions, we would appreciate it if you could kindly consider raising the score.
>
> Sincerely yours,
>
> Paper5095 Authors

---

### Author Response · Authors · 2022-07-27
**General Response 0: To All Reviewers**

Dear Reviewers,

We appreciate your effort and time in reviewing our paper. We have noticed that some parts of Sec. 4.1 and 4.2 may not easy for you to follow. The main reason is the following part should be inserted into Line 150 before the sentence **Unlike many parameterized episodic memory using state/observation ...**. We mistakenly deleted that part during the main text submission phase. This part can also be found in Appendix B.1 in the supplementary file.

> Besides the $\phi^t_i$, we define the sub-graph set of $\phi^t_i$ as $\Phi^{t, \Omega}_i=\\{\phi^{t,\omega}_i\\}^\{\Omega-1\}_\{\omega=0\}$ by using the discretized episode return and there are $\Omega$ sub-graphs. $\phi ^ {t, \omega}_i$ is the $\omega$-th sub-graph whose episode return is $\Upsilon \texttt{[} \omega \texttt{]}$ ($\omega \in \\{0, \cdots, \Omega-1\\}$, $\Upsilon=\[0, \cdots, \boldsymbol{r}^{t,i}\]$) where $\boldsymbol{r}^\{t,i\}$ is the discretized maximum episode return of $\phi^t_i$.

We believe adding this part can help you better understand our paper. In the final version, we will put the above paragraph in the main text. More discussions on our paper are always welcomed!


Best regards,

Paper5095 Authors

---

### Author Response · Authors · 2022-07-30
**General response 1: The Flow of Our Paper**

Dear reviewers, to address your concerns and enhance your understanding of our paper, we would like to introduce the flow of our paper in the following. We hope our response can address your concerns. More questions and discussions are welcomed.

We introduced the off-beat problem in the abstract (Lines 1-5) and then proposed our solution. In the Introduction section (Sec. 1), we introduced the background in paragraph 1 and introduced the problem of off-beat actions with an intuitive example in Fig. 1 (page 2) in paragraph 2. We introduced the challenges in off-beat multi-agent scenarios (OBMAS) in paragraph 3. We discussed the drawbacks and issues of existing works in paragraph 4. In paragraph 5, we summarized our contributions.

Readers can find the preliminaries in Sec. 2. Related works on Action Delay in RL, Credit Assignment in RL and Multi-Agent RL are in Sec. 7, page 9.

After reading the abstract, the introduction and the example in Fig. 1, we believe readers have an intuitive understanding of our problem. So, in Sec. 3, we formally introduced Off-Beat Action and Off-Beat Dec-POMDP together with four remarks on the properties of Off-Beat Dec-POMDP.

Our paper aimed to propose a solution for learning cooperative multi-agent RL in off-beat multi-agent scenarios. Inspired by **mental time travel time** and **episodic memory works** (see Lines 122-128, in Sec. 4.1, page 4), we propose our episodic memory, called LeGEM (see Sec. 4.1, pages 4-5). LeGEM is built by utilizing the experiences in the repay buffer. LeGEM builds episodic memory by structuring agents’ experiences. The reason why we use the title **“The Journey is the Reward: A Collective Mental Time Travel Method”** for Sec. 4 is that although the action duration is not accessible and the observation is partial observable, the **pattern** of “Action-Reward Association” exists ( Fact 1, in Line 167, page 5, Sec. 4.2). So, every trajectory (journey) matters. LeGEM can be considered an implicit environment model, and it can be used to search the pivot timesteps.

We aim to learn cooperative MARL in off-beat multi-agent scenarios (OBMAS). However, the rewards are displaced, rendering the credit assignment issue in TD learning in MARL. Luckily, **the reward is the key to the learning in RL and MARL**. Inspired by this philosophy, we utilize LeGEM to search the pivot time steps (see Sec. 4, pages 5-6) and redistribute the reward to the pivot time step in Sec. 5, pages 6-7.

Learning cooperative MARL is a critical problem, and it is the same case in Off-Beat MARL. We evaluate our method in Sec. 6. Experimental results show that our method can learn good cooperative policies in OBMAS.

We made closing remarks by summarizing our paper and discussing limitations and future work in Sec. 8, page 9.

Best regards,

Paper5095 Authors

---

### Author Response · Authors · 2022-08-02
**General response 2: We added the Results of MARL with RUDDER**

Dear Reviewers,

We updated the Appendix with the results of MARL with RUDDER. Changes in the Appendix are in blue. You can find the results in Sec. F.2 in the Appendix. To sum up, MARL with RUDDER performs poorly in OBMAS.

On Stag-Hunter Game, Quarry Game, Afforestation Game, and SMAC scenario 2m_vs_1z, QMIX-RUDDER, VDN-RUDDER, IQL-RUDDER and QTRAN-RUDDER all perform poorly, as shown in Fig. 18, 19, 20 and 21. The performances are all zero. There are three reasons caused the poor performance:

(i) RUDDER cannot capture the association between off-beat actions and off-beat rewards, making it challenging to detect the pivot timesteps and redistribute the episodic reward to pivot timesteps;

(ii) RUDDER conducts the contribution analysis by estimating the reward of each timestep via regression. Due to the sparsity of off-beat rewards and the estimation error of RUDDER, RUDDER redistributes the reward to timesteps around the pivot timestep, rendering the failure of TD learning;

(iii) Our setting is a partial observable multi-agent setting, simply redistributing rewards without considering the multi-agent setting can redistribute the reward to the wrong time steps. In TD learning, estimating the TD target is essential to learning the policy (or the Q value). However, with off-beat actions, n-step return and TD(λ) fail in Off-Beat MARL, as shown in Table 2 in the main text and Sec. F.1 in Appendix. It is not surprising to see that RUDDER also fails in Off-Beat MARL.

---

We hope the updated results of MARL with RUDDER can address your concerns. More questions are always welcomed!

Best regards,

Paper5095 Authors

---

### Author Response · Authors · 2022-08-03
**General response 3: Dear reviewers, did our responses address your questions?**

Dear Reviewers,

We thank all the reviewers again for their constructive and valuable comments. We appreciate the positive comments by reviewers who recognised our contribution to MARL.

We hope our responses, including experiments of RUDER, could address the reviewers' questions. More discussions and suggestions on our paper are also always welcomed!


Sincerely yours,

Paper5095 Authors

---

### Author Response · Authors · 2022-08-08
**General response 4: We updated our Paper**

Dear Reviewers,

We are glad to hear that our responses addressed some of your concerns. As suggested by Reviewer CAwi, we incorporated a more intuitive explanation of the graph search algorithm for non-experts readers. The main updates are in Appendix B.4 (in blue), and we also updated the main text in Sec. 5 (in blue). As the response system will be closed soon within less than 2 days, if you have more questions about our paper, please let us know. We are looking forward to your opinions and further discussions on our paper.


Sincerely yours,

Paper5095 Authors

---

### Meta-Review · Area_Chair_GGmN · 2022-08-26

**Recommendation:** Reject
**Confidence:** Less certain

**Metareview:**

The review scores for this paper were borderline, and while close to the bar for acceptance, I am unable to recommend acceptance at this time, due to the very competitive nature of NeurIPS submissions. Reviewers had mixed opinions on the applicability of the off-beat scenario presented, and concerns about missing baselines (especially using methods from single-agent RL that are better able to handle delayed effects, e.g. alternative memory architectures), clarity of presentation, and scalability. Personally, I find the problem setting interesting, but am most concerned about the aforementioned baselines. All of that said, I think this work is promising and encourage the authors to integrate reviewer feedback and resubmit.

**Award:**

No

---

### Decision · Program_Chairs · 2022-09-14

Reject